# Antimicrobial Activities against Opportunistic Pathogenic Bacteria Using Green Synthesized Silver Nanoparticles in Plant and Lichen Enzyme-Assisted Extracts

**DOI:** 10.3390/plants11141833

**Published:** 2022-07-13

**Authors:** Aistė Balčiūnaitienė, Paulina Štreimikytė, Viktorija Puzerytė, Jonas Viškelis, Žaneta Štreimikytė-Mockeliūnė, Žaneta Maželienė, Vaidė Sakalauskienė, Pranas Viškelis

**Affiliations:** 1Lithuanian Research Centre for Agriculture and Forestry, Institute of Horticulture, 54333 Babtai, Lithuania; paulina.streimikyte@lammc.lt (P.Š.); viktorija.puzeryte@lammc.lt (V.P.); jonas.viskelis@lammc.lt (J.V.); pranas.viskelis@lammc.lt (P.V.); 2Institute of Microbiology and Virology, Lithuanian University of Health Sciences, Eivenių g. 2, 50161 Kaunas, Lithuania; zaneta.streimikyte@lsmuni.lt (Ž.Š.-M.); zaneta.mazeliene@lsmu.lt (Ž.M.); 3Mėlynė JSC, Sodų Str. 5, 54334 Babtai, Lithuania; vaide.sakalauskiene@mita.lt

**Keywords:** green synthesis, silver nanoparticles, *Cetraria islandica*, *Fagopyrum esculentum*, enzyme-assisted extraction, *Trichoderma reesei*

## Abstract

Enzyme-assisted extraction is a valuable tool for mild and environmentally-friendly extraction conditions to release bioactive compounds and sugars, essential for silver nanoparticle (AgNP) green synthesis as capping and reducing agents. In this research, plant and fungal kingdoms were selected to obtain the enzyme-assisted extracts, using green synthesized AgNPs. For the synthesis, pseudo-cereal *Fagopyrum esculentum* (*F. esculentum*) and lichen *Certaria islandica* (*C. islandica*) extracts were used as environmentally-friendly agents under heating in an aqueous solution. Raw and enzyme-assisted extracts of AgNPs were characterized by physicochemical, phytochemical, and morphological characteristics through scanning and transmission electron microscopy (SEM and TEM), as well as Fourier transform infrared spectroscopy (FTIR). The synthesized nanoparticles were spherical in shape and well dispersed, with average sizes ranging from 10 to 50 nm. This study determined the total phenolic content (TPC) and in vitro antioxidant activity in both materials by applying standard methods. The results showed that TPC, ABTS^•+^, FRAP, and DPPH^•^ radical scavenging activities varied greatly in samples. The AgNPs derived from enzymatic hydrolyzed aqueous extracts *C. islandica* and *F. esculentum* exhibited higher antibacterial activity against the tested bacterial pathogens than their respective crude extracts. Results indicate that the extracts’ biomolecules covering the AgNPs may enhance the biological activity of silver nanoparticles and enzyme assistance as a sustainable additive to technological processes to achieve higher yields and necessary media components.

## 1. Introduction

Nanoparticles (NPs) are helpful in material science, medicine, pharmaceutical, food processing, biotechnology, agriculture, environment, and energy. The unique properties of NPs have attracted current researchers owing to their novel biological, chemical, and optical properties [1]. Silver nanoparticles (AgNPs) are predominant in biomedical and medical applications [2]. AgNPs show strong efficacy against bacteria, fungi, viruses, and other microorganisms and proved to be effective health additives in medicine and pharmacy. It is relevant to develop an environmentally-friendly NP synthesis method that does not use toxic chemicals and protects natural resources in the synthesis processes [3]. Green chemistry emphasizes deploying principles that reduce or eliminate the use or generation of hazardous substances in chemical reactions [4]. Among the essential aims of green chemistry is environmental protection. Biological methods such as green synthesis are described as non-toxic, simple, rapid, and dependable among several reported methods. Green methods can generate AgNP’s well-defined size and morphology under optimized conditions and reproducibility compared to the physical and chemical methods [2,5,6]. The intrinsic ability of plants and their secondary metabolites and sugars, to act as reducing and capping agents and contribute to the amalgamation of metal ions into nanoparticles is more reputed than that of microorganisms (bacteria and fungi). Further, using plant extracts to synthesize silver nanoparticles is much faster than routes involving microorganisms. Most commonly found secondary metabolites, such as flavonoids, phenolics, tannins, terpenoids, and alkaloids, mediate the green synthesis of AgNPs at a molecular level, thereby contributing to their antimicrobial activities [7,8].

Lichens are composite organisms consisting of algae and fungi or cyanobacteria in symbiotic associations. Lichens belong to the fungal kingdom and can be found almost anywhere in the world since they can withstand even the harshest ecological conditions [9,10,11]. Lichens grow on soils, trees, and rocks; plants’ morphology is crusty, leafy, and branched. *C. islandica* is a lichen species most commonly found in damp places, tundra, bogs, and conifer-hardwood trees and rocks [12]. *C. islandica* is composed of bitter lichen acids such as cetraris and usnic acids in addition to 50% polysaccharides–isolichenin, lichenin, and naphtoquinones, such as naphtazarin, minerals, other B vitamins, and folic acid [13]. Lichenin is found in thallus and used to isolate β-glucans, which, if enzymatically hydrolyzed, release monomeric sugars suitable for AgNP’s synthesis [13]. However, these species are better known for high antioxidant activity, reducing the force, with peroxide anion radical removal and free radical removal activities [13,14]. *C. islandica* is widely used in folk medicine for fever, lung and kidney diseases, throat and mouth problems, bronchitis and dry cough, and susceptibility to infection [15,16].

Another good plant is common buckwheat *F. esculentum*, an essential short-season crop worldwide. *F. esculentum* is widely grown in Europe, America, and Asia and has proven to be a portion of healthy food, containing large amounts of macro and micro components, functional polysaccharides, and bioactive compounds [17,18]. Buckwheat contains a high level of starch, which, if hydrolyzed, may play a key role in synthesizing nanoparticles as a reductor. Buckwheat starch contains amylose of concentration in the range of 15–52% with polymerization degrees from 12 to 45 glucose units. *F. esculentum* polysaccharides potentially initiate inhibition response against immunosuppressor CTX in mice’s. From a bioactive composition perspective, buckwheat has a higher antioxidative (tocopherols and polyphenols) activity and phenolic content than other cereal and pseudo-cereal grains [19].

Green synthesis media must be enriched with bioactive composition. During extraction, enzymatic hydrolysis of carbohydrates greatly enhances phenolic content by making the cell wall permeable. Moreover, non-starch polysaccharide enzymes (cellulases, xylanases, β-glucanases) with amylolytic enzymes convert long-chain carbohydrates into soluble reducing sugars, resulting in their ability to act as reducing agents under specific conditions [20]. In general, carbohydrates are the most potent agents disturbing the complex matrix of plant material, which releases bioactive compounds into aqueous extracts [21,22,23]. 

The present scientific study focused on comparing the biosynthesis of AgNPs of raw and enzyme-assisted extracts of pseudo-cereal *F. esculentum* and lichen *C. islandica* and evaluating antimicrobial activity.

## 2. Materials and Methods

### 2.1. Lichen and Plant Materials

The lichen *Cetraria islandica* was collected in Sakiai District Municipality in the forest of Zapyskis, Lithuania. Lichen material was ground to a powder using a mill (IKA^®^ A11 basic “Staufen”, Germany). Before analysis, loss on drying was determined by drying powdered raw lichen material in a laboratory drying oven to complete evaporation of water and volatile compounds (temperature: 105 °C; the difference in weight between measurements: up to 0.01 g) and by calculating the difference in raw material weight before and after drying. The data were recalculated for absolute dry weight.

Roasted common buckwheat flour (*F. esculentum* L.) material was obtained from the local Lithuanian producers (Producer “Ekofrisa”, Prienai country, Naraukelis, Lithuania). Plant material was ground by an ultra-centrifugal mill ZM 200 (Retsch, Haan, Germany) using a 0.5 mm hole size sieve before extraction.

### 2.2. Enzyme-Assisted Extraction

An amount of 20 g of raw material of each plant is weighed (exact mass of ground material), poured over 120 g of distilled, and added to the 0.15% (*w*/*w*) the multienzyme non-starch polysaccharide (NSP) cocktail (JSC “Baltijos Enzimai”, Lithuania) as described by Streimikyte et al. [24]. The main enzyme cocktail component is cellulase. The product is declared to have a 5000 U/mL activity of cellulase. Additionally, the manufacturer states that the enzyme cocktail contains various hemicellulose, endo-xylanase, exo-xylanase, beta-glucanase, mannanase, galactosidase, and pectinase activities. Enzymes were produced from *Trichoderma reseii*. Extraction was carried out for 2.5 h at a temperature of 65 °C; the obtained extracts were separated and centrifuged. Prepared lichen and plants ferment, and raw extracts were used as green reductants and capping agents for the biosynthesis of AgNPs.

### 2.3. Preparation of AgNPs

Preparation of the samples was implemented as described by Balciunaitiene et al. [25]. Silver nitrate (AgNO_3_) (Merck, Darmstadt, Germany) was weighed 0.03 g and dissolved in 3 mL of distilled water. A 30 mL of each aqueous raw and fermented extracts of *C. islandica* and *F. esculentum* were added to AgNO_3_ aqueous solution under vigorous stirring (speed 500 rpm) at 45 °C temperature for 2 h using magnetic stirrer (IKA^®^ C-MAG MS, Germany), after that incubation at a room temperature for 24 h. During synthesis, a gradual change in color from a mild yellow to a dark brown in the reaction mixture indicates the presence of silver nanoparticles

### 2.4. Spectrophotometric Studies

#### 2.4.1. Color Measurement and Active Acidity (pH)

The extracts and solutions of the formed AgNPs were evaluated according to the color coordinates, which confirmed a silver nanoparticle synthesis. Samples were measured by a spectrophotometer MiniScan XE Plus (Hunter Associates Laboratory Inc., Reston, VA, USA) following the method described in Urbonaviciene’s et al. paper [26,27]. The color parameters were processed with the software Universal V.4–10. Active acidity (pH) was measured with inoLab pH Level 1 pH meter using SenTix 81 (WTW, London, UK) electrode. Measurements were performed in triplicate.

#### 2.4.2. Fourier Transform Infrared (FTIR)

Fourier transform infrared (FTIR) analysis was performed using a Spectrum GX Perkin-Elmer FTIR spectrometer (Waltham, MA, USA), and the method attenuated total reflection was applied. The FTIR spectra were collected in the frequency range of 4000–650 cm^−1^, and ten scans were performed for each tested sample [28].

### 2.5. Microscopy

The size and structure of AgNPs were studied from images obtained using a transmission electron microscope (TEM) Tecnai G2 F20 X-TWIN (FEI, Lausanne, Switzerland), equipped with a field emission electron gun at the accelerating voltage of 200 kV. For a microscopy study, the diluted samples were deposited on the TEM grids. AgNP diameter was measured with software ImageJ-win32 [7,29,30]. In addition, the microstructure of the sample was analyzed using a Scanning Electron Microscope SEM FEI Quanta 200 FEG (FEI, Hillsboro, OR, USA). The plant and lichen samples were examined in a low-vacuum mode operating at 20.0 kV using an LDF detector.

### 2.6. Analysis of Total Phenolic Compounds

The total polyphenol content in the extracts was determined according to the Folin–Ciocalteu method, using gallic acid (GA) as the standard, according to the method of Bobinaite et al. [31].

### 2.7. In Vitro Antioxidant Activity Assays

The radical scavenging activity (RSA) of the extracts against the stable DPPH^•^ was determined using the slightly modified spectrophotometric method described by Urbonaviciene et al. [32].

The ABTS^•+^ assay was performed, as described by Re et al. [33]. The ferric-reducing activity (FRAP) was determined, according to the method of Benzie and Strain [34], with some modifications, according to Raudone et al. [35]. All antioxidant activity measurements and calculations were performed using Trolox calibration curves and were expressed as µmol of the Trolox equivalent (TE) per one gram of dry weight.

### 2.8. Antimicrobial Activity

The antimicrobial activity of the extracts and synthesized AgNPs were investigated against Gram-negative and Gram-positive bacteria cultures. The agar diffusion and minimal inhibitory concentration (MIC) tests were chosen to evaluate antibacterial activity. The antimicrobial activity of extracts was tested via an agar well diffusion assay. For this purpose, a 0.5 McFarland Unit density suspension (~10^8^ CFU mL^−1^) of each pathogenic bacterial strain was inoculated onto the surface of solid Mueller Hinton Agar (Oxoid, Basingstoke, UK) using sterile cotton swabs. Wells of 6 mm diameter were punched in the agar and filled with 50 µL of extracts. The experiments were repeated three times, and the average size of the inhibition zones was calculated. The antimicrobial activities against the tested bacteria were determined by measuring the diameter of the inhibition zones (mm).

MIC was defined as the lowest extracts concentration inhibiting visible microbial growth. Experiments were performed in triplicate.

The antimicrobial activity of extracts was determined against *Staphylococcus aureus* ATCC 25923, *beta hemolytic streptococcus* group b ATCC 15185, *Staphylococcus epidermidis* ATCC 12228, *Enterococcus faecalis* ATCC 29212, *Escherichia coli* ATCC 25922, *Klebsiella pneumoniae* ATCC 13883, *Pseudomonas aeruginosa* ATCC 27853, *Proteus vulgaris* ATCC 8427, *Bacillus cereus* ATCC 11778, *Listeria monocytogenes* ATCC 19115, and *Candida albicans* ATCC 10231 Lithuanian University of Health Sciences (Kaunas, Lithuania).

### 2.9. Statistical Analysis

The statistical analysis was performed using SPSS 20 software (SPSS Inc., Chicago, IL, USA). Significant differences between the means of the total phenolic content and in vitro antioxidant capacities were evaluated using ANOVA and post hoc Tukey’s HSD multiple comparison test. All the data were prepared in triplicate (*n* = 3), and the results are shown as the mean ± standard error mean (SEM) of all calculated values.

## 3. Results and Discussion

### 3.1. Color Measurement, Active Acidity and Sugars

The positive outcome in the green synthesis of AgNPs using the lichen *C. islandica* and plant *F. esculentum* enzyme-assisted extracts was confirmed by a color change and spectroscopic study of the reaction medium and nanoparticles. There was an apparent color change from a light cream to brown, and a significant difference was observed after a mixture of plant and lichen extracts and silver nitrate (see Table 1). With these changes in color, the conversion of ionic Ag^+^ to metallic Ag^0^ that self-ensembles into colloidal particles was suggested. This observation was corroborated by those obtained by other authors; Kambale et al., 2020 and Jain et al., 2017 [7,29].

pH, sugars, and phenolic content are essential to the green synthesis of nanoparticles. After enzymatic hydrolysis, sugars as reductors were released into extracts [36]. However, *F. esculentum* sugar content in the extract was 6.8-fold higher than *C. islandica*. Moreover, pH is vital to synthesis, and active acidity was decreased in both *C. islandica* and *F. esculentum* samples. As seen from the presented data in Table 1, in both cases, pH decreased slightly after the synthesis of AgNPs.

### 3.2. Morphological Analysis

#### 3.2.1. Scanning Electron Microscopy (SEM)

After enzyme-assisted extraction, the solid fractions of *C. islandica* and *F. esculentum* were assessed using SEM to evaluate the enzyme disturbance of cell walls and applicability (Figure 1). Cellulolytic enzymes such as cellulose and xylanase play a crucial role in the disruption of polysaccharides of lichen and buckwheat. By cleavage of long-chain cellulose, hemicellulose, and lignin, forces and charges of the surface changed [36]. Interestingly, depending on subtract origin, enzymatic cellulase derivatives act differently due to the surface’s charge orientation [37,38]. The overview and detailed appearance of the *C. islandica* and *F. esculentum* solid fractions used before and after fermentation are shown in Figure 1. *C. islandica* particles’ surface has higher roughness and has a irregular geometric shape (Figure 1a,b). Moreover, after enzyme-assisted extraction, microfibrils and amorphous zones can be seen at all particle locations. These areas have increased, and this should also increase the extraction yield. After enzymatic hydrolysis of *F. esculentum* (Figure 1c,d), particles can be seen in micro-size agglomeration with an octagonal and are closely aligned.

Specifically, exo-1,4-β-cellulase and β-glucosidase are synergically active during enzyme-assisted water extraction by releasing the free-reducing and non-reducing ends of the glucose chains. Disruption of cells also increased the yield of extract as well as reduced sugars and phenolic content, which are involved in the synthesis of silver nanoparticles [8,39].

#### 3.2.2. Transmission Electron Microscopy (TEM)

After AgNP synthesis in plant and lichen extracts was performed, TEM was used for its significant application in studying nanoparticle shape and size [40]. The microscopic observation of the green-synthetic reaction medium allowed verifying the formation of colloidal AgNPs, as shown in Figure 2. The particle size and the distribution of the synthesized AgNPs were estimated under TEM using ImageJ software. Particle size was found to be approximately <50 nm.

As shown, the obtained nanoparticles have irregular geometric shapes but are primarily spherical and do not tend to form large aggregations; this is inherent in silver nanoparticles [41]. Moreover, plant and lichen extracts are involved in the capping function [40].

Further, all the ImageJ graphs described a Gaussian distribution peak at approximately the average sizes estimated for each nanoparticle formulation (Figure 2), which shows acceptable particle size distribution as observed with polydispersity values below 50%.

As shown in Figure 3, EDX analysis revealed the presence of AgNPs formulated from *C. islandica* and *F. esculentum* extracts. As widely known, the optical absorption peak of Ag appears at approximately 3 keV due to surface plasmon resonance [42].

While Ag peaks were the most prominent for tested samples, the EDX patterns of the prepared formulations did not show any nitrogen peaks. Results indicate the absence of noticeable traces of ions from AgNO_3_ initially used. This confirms the presence of metallic silver in the sample and is evidence of the successful reduction of Ag ions.

### 3.3. Fourier Transform Infrared (FTIR)

Figure 4 presents the FTIR spectra of the AgNPs synthesized using control and enzyme-assisted extracts of *C. islandica* and *F. esculentum*. From all the FTIR spectra, some stretching vibration bands related to organic functional groups were prominently observed. *C. islandica* is present in round-shaped bands at 1050–1025 and 1380–1360 cm^−1^ (C–O phenolic and O–H stretches). Further, stretches at 2917–2833 cm^−1^ are alkanes, C-H [29,42,43]. The stretching mode was observed at 1640–1620 cm^−1^ (C=O), detecting aldehydes, carboxylic, ketone, or ester-containing compounds [44].

It is important to note that FTIR spectra did not detect the nitro bond at 1280 cm^−1^ of the green synthesized AgNPs and this confirms the absence of noticeable traces of nitrates. These data approve of the successful formation of the silver nanoparticles, which were obtained by green synthesis by reducing phytomolecules—secondary metabolites.

### 3.4. Total Phenolic Content and In Vitro Antioxidant Activity

Total phenolic content (TPC) and three different in vitro radical scavenging activities of enzyme-assisted extracts with and without silver nanoparticles of *C. islandica* and *F. esculentum* were measured [45]. In general, there is much interest in secondary metabolites, including phenolic compounds, as a result of their antioxidant abilities that lead to significant improvement in human health [46]. Moreover, for synthesis, a higher content of TPC may increase the active surface area of AgNPs as capping agents [47]. 

The results in Table 2 indicate the TPC content of *F. esculentum* and *C. islandica* enzyme-assisted water extracts with and without synthesized AgNPs. For buckwheat and lichen, they reached 37.77 and 14.53 mg GAE/100 mL, respectively. Moreover, Table 2 shows that hydrolyzed and released components have radical antioxidant scavenging capacities. *F. esculentum*/*EAE*/*AgNPs* and *C. islandica*/*EAE*/*AgNPs* showed the highest activity against DPPH^•^—5.20 ± 0.09 and 2.99 ± 0.10 µM TE/100 mL, respectively—although all samples stand out with a significant difference, except FRAP radical scavenging activity. *F. esculentum* showed higher phenolic content and antioxidant activity in enzyme-assisted aqueous extracts.

However, most of the lichen’s secondary metabolites are extractable with organic solvents. Structurally, lichen and buckwheat consist of different amounts of carbohydrates. In buckwheat polysaccharides, where starch is the main component, in order for *C. islandica*, thallus structure is approximately 50% of lichenin and isolichenin [15,48].

### 3.5. Antimicrobial Activity

The antimicrobial activity of silver nanoparticles against many species of bacteria is well known and widely used [7,28,49,50,51]. Our tests performed against Gram-positive and Gram-negative bacteria and fungi using disc diffusion or the Kirby–Bauer and MIC methods containing different types of extracts showed significant changes. The majority of inhibitory zones, which were observed, confirmed the diffusion of silver nanoparticles from extracts into the medium. The results of the extracts for microbiological tests are presented in Table 3. The antimicrobial activity of the pure extracts shows biocidal activity only in *C. islandica*. These results corroborate with those obtained by Meli et al. [15].

Compared with pure extracts, results shown in Table 3 and Table 4 show the increasing antimicrobial activity of bacterial diversity in enzyme-assisted extracts. It is conceivable that amplified biological active components, dispersed in the aqueous extracts during enzymatic hydrolysis, correlated with progressive biocide activity.

However, the synthesized AgNPs in pure extracts exhibited remarkable antibacterial activities, which were found to be greater than those of *C. islandica*/*EAE* and *F. esculentum*/*EAE*. The *C. islandica*/*EAE*/*AgNPs* actively inhibit the bacterial test strains, and in the case of *S. aureus* and *ß-streptococcus*, the size of the inhibition zone is as large as 19.5 mm. Similar activity was obtained for *F. esculentum*/*EAE*/*AgNPs*, where the size of the inhibition zone was ~13 mm.

Antibacterial effects of AgNPs were investigated against Gram-positive and Gram-negative bacteria and fungi through determination of the MICs.

Pure extracts show weak antimicrobial activity. However, the AgNPs from *C. islandica/EAE* exhibited remarkable antibacterial activities, which were greater than AgNP synthesis from *F. esculentum*.

The antimicrobial activity of AgNPs is solely due to Ag^+^ release, and even a relatively low concentration of silver ions accounts for the biological response. This could be explained by the fact that particular phytoactive compounds covered the AgNPs from extracts (see the 2500–1750 cm^−1^ region of the FTIR spectra, Figure 4). These phytochemicals that may act as reducing and/or capping agents would be preferentially involved in the reduction of Ag^+^ ions into Ag^0^ metal and in the phytochemical-assisted fabrication of AgNPs from *C. islandica* and *F. esculentum* extracts (nucleation, capping and stabilization).

Furthermore, an experimental design has shown that multienzyme-assisted water extracts of *C. islandica* and *F. esculentum* compared with pure extracts have 2-foldhigher antibacterial effects against Gram-positive bacteria than Gram-negative bacterial cultures. 

The antimicrobial activity of AgNPs in Gram-positive and Gram-negative bacterial cultures may differ due to differences in bacterial cell structure. The cell wall of Gram-positive bacteria is 15–80 nm thick and consists of several dozen layers of peptidoglycan. This wall is not dense, so even large molecules can pass through. As a result, most biologically active substances can quickly enter the cell of these bacteria and affect their growth and reproduction. Contrastingly, Gram-negative bacteria are less sensitive to antibacterial compounds because the cell wall has one or more layers of peptidoglycan and an additional outer membrane composed of lipopolysaccharides, lipoproteins, and phospholipids [49].

## 4. Conclusions

The present study demonstrated the green and eco-friendly fast, facile, and economical synthesis of AgNPs using aqueous and enzyme-assisted extraction of *Cetraria islandica* and *Fagopyrum esculentum* frequently used in traditional food industry and medicine. Different test methods confirmed the successful formation of spherical, individual silver nanoparticles. They evidenced the role of phytochemicals from the extracts and the enzymatic hydrolysis influence of reducing and capping agents in the green synthesis of silver nanoparticles. All the synthesized AgNPs using the green method displayed comparable antioxidant activity. Moreover, in contrast to *C. islandica* samples, total phenolic content in common buckwheat after and before silver synthesis was significant difference. Additionally, the antimicrobial evaluation of AgNPs revealed excellent antibacterial activity against all tested Gram-positive and Gram-negative bacterial strains and fungi, which are hazardous pathogens commonly involved in infectious diseases. The obtained results in our study contribute to a novel and environmentally-friendly area of a synthesis method and nanomaterial as a promising method in medicine and pharmacy. Moreover, future studies are needed to fully characterize the influence of enzymes on AgNP toxicity and stability. 

## Figures and Tables

**Figure 1 plants-11-01833-f001:**
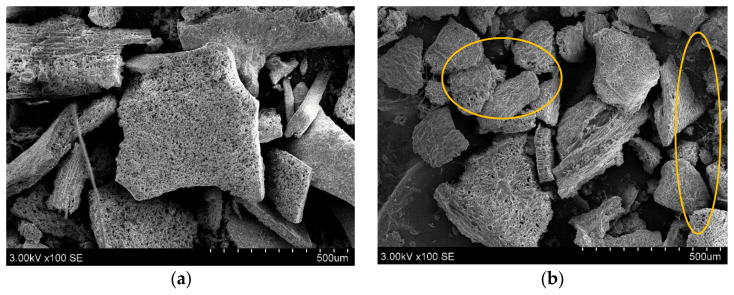
SEM images of *C. islandica* (**a**,**b**) and *F. esculentum* (**c**,**d**) solid fractions before and after enzymatic hydrolysis.

**Figure 2 plants-11-01833-f002:**
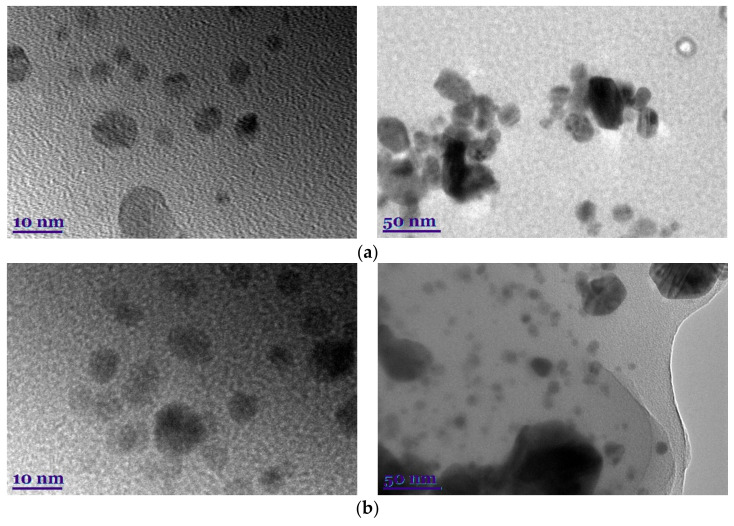
TEM micrographs of AgNPs using *C. islandica* (**a**), *F. esculentum* (**c**), and enzyme-assisted *C. islandica* (**b**) and *F. esculentum* (**d**) extracts.

**Figure 3 plants-11-01833-f003:**
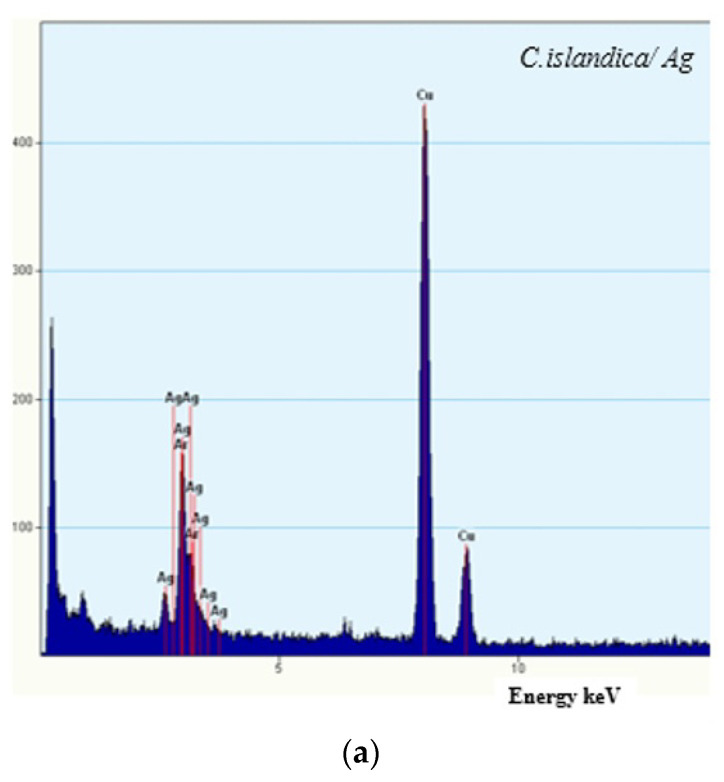
EDX spectra of green synthesized AgNPs using enzyme-assisted *C. islandica* (**a**) and *F. esculentum* (**b**) extracts.

**Figure 4 plants-11-01833-f004:**
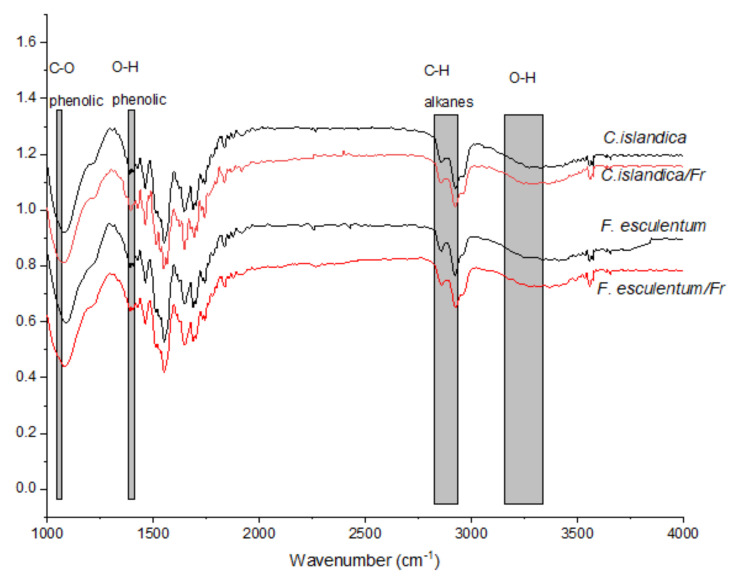
FTIR spectra of the bio-reduced AgNPs using enzyme-assisted (in Figure 1 labeled as *F. esculentrum*/*Fr and C. islandica*/*Fr*) and control (in Figure 1 labeled as *F. esculentrum and C. islandica*) *C. islandica* and *F. esculentum* extracts.

**Table 1 plants-11-01833-t001:** Color coordinates, pH and sugar content in *C. islandica* and *F. esculentum*.

Sample	L *	A *	B *	pH	Sugars g/mL
*C. islandica*	47.45 ± 0.17 ^f^*	0.98 ± 0.04 j	9.83 ± 0.18 ^h^	4.9	0.0013 ± 0.0002
*C. islandica*/*Ag*/*NPs*	26.72 ± 0.15 ^g^	11.32 ± 0.09 ^k^	7.02 ± 0.33 ^i^	4.44	-
*F. esculentum*	46.13 ± 0.07 ^a^	0.32 ± 0.07 ^c^	4.71 ± 0.10 ^ed^	6.7	0.0089 ± 0.0003
*F. esculentum*/*Ag*/*NPs*	32.71 ± 0.12 ^b^	3.37 ± 0.17 ^d^	5.93 ± 0.33 ^e^	6.05	-

* Different letters in the same row indicate a significant difference between the measured values (*p* ≤ 0.05).

**Table 2 plants-11-01833-t002:** Total phenolic content and in vitro antioxidant capacity of *F. esculentum* and *C. islandica*.

	TPC, mg GAE/100 mL	ABTS^•+^, µM TE/100 mL	DPPH^•^, µM TE/100 mL	FRAP, µM TE/100 mL
*F. esculentum*	42.00 ± 1.66 ^a^	5.66 ± 0.11 ^a^	7.43 ± 0.10 ^a^	3.12 ± 0.14 ^a^
*F. esculentum*/*EAE*/*AgNPs*	37.77 ± 1.07 ^b^	4.96 ± 0.07 ^b^	5.20 ± 0.09 ^b^	2.73 ± 0.53 ^ab^
*C. islandica*	15.63 ± 1.40 ^c^	2.50 ± 0.19 ^c^	4.45 ± 0.30 ^c^	2.30 ± 0.11 ^cb^
*C. islandica*/*EAE*/*AgNPs*	14.53 ± 0.90 ^c^	2.22 ± 0.09 ^d^	2.99 ± 0.10 ^d^	1.92 ± 0.12 ^c^

Different letters in the same row indicate a significant difference between the measured values (*p* ≤ 0.05). EAE—enzyme-assisted extract.

**Table 3 plants-11-01833-t003:** Antimicrobial activity of the green synthesized AgNPs.

Reference (Standard) Cultures of Microorganisms	Samples							
	1	2	3	4	5	6	7	8
	Units, mm							
*Staphylococcus aureus*	1.5 ± 0.1	9.1 ± 0.1	8.7 ± 0.1	11.40 ± 0.7	0.0 ± 0.0	0.5 ± 0.1	13.5 ± 0.5	15.6 ± 0.0
*Staphylococcus epidermidis*	1.4 ± 0.2	7.8 ± 0.1	5.9 ± 0.1	16.5 ± 0.2	0.0 ± 0.0	0.1 ± 0.5	12.5 ± 0.5	13.0 ± 0.0
*ß-streptococcus*	1.7 ± 0.1	5.9 ± 0.1	16.5 ± 0.2	18.2 ± 0.7	0.0 ± 0.0	0.6 ± 0.2	10.8 ± 0.0	16.1 ± 0.4
*Escherichia coli*	2.5 ± 0.1	2.8 ± 0.3	15.1 ± 0.1	16.7 ± 0.1	0.0 ± 0.0	1.7 ± 0.6	8.6 ± 0.3	10.1 ± 0.0
*Klebsiella pneumoniae*	1.0 ± 0.5	2.2 ± 0.6	12.4 ± 0.2	13.0 ± 0.1	0.0 ± 0.0	2.3 ± 0.5	9.7± 0.0	10.4 ± 0.0
*Pseudomonas aeruginosa*	1.2 ± 0.2	2.7 ± 0.7	10.3 ± 0.5	15.7 ± 0.3	0.0 ± 0.0	1.6 ± 0.1	9.0 ± 0.1	11.2 ± 0.5
*Proteus vulgaris*	2.4 ± 0.1	2.4 ± 0.4	14.4 ± 0.7	15.8 ± 0.2	0.0 ± 0.0	0.9 ± 0.2	7.5 ± 0.5	9.4 ± 0.3
*Bacillus cereus*	1.5 ± 0.4	8.7 ± 0.3	8.2 ± 0.4	9.8 ± 0.1	0.0 ± 0.0	0.5 ± 0.1	12.8 ± 0.2	14.7 ± 0.1
*Enterococcus faecalis*	1.2 ± 0.3	6.4 ± 0.2	9.0 ± 0.1	10.7 ± 0.2	0.0 ± 0.0	0.4 ± 0.2	14.0 ± 0.1	15.7 ± 0.2
*Candida albicans*	0.4 ± 0.4	1.0 ± 0.1	8.1 ± 0.3	10.0 ± 0.1	0.0 ± 0.0	0.4 ± 0.8	6.5 ± 0.2	7.8 ± 0.7

1. *C. islandica*; 2. *C. islandica*/EAE 3. *C. islandica*/AgNPs; 4. *C. islandica*/EAE/AgNPs; 5. *F. esculentum*; 6. *F. esculentum*/EAE; 7. *F. esculentum*/AgNPs; 8. *F. esculentum*/EAE/AgNPs. EAE—multienzyme-assisted extract.

**Table 4 plants-11-01833-t004:** Minimal inhibitory concentrations against pathogenic bacteria.

	Reference (Standard) Cultures of Microorganisms									
Samples	*Staphylococcus aureus*	*Staphylococcus epidermidis*	*Enterococcus faecalis*	*Escherichia coli*	*Klebsiella pneumoniae*	*Pseudomonas aeruginosa*	*Proteus vulgaris*	*Bacillus cereus*	*Listeria monocytogenes*	*Candida albicans*
	µL/mL									
*1*	166.67	-	166.67	-	-	-	-	166.67	166.67	-
*2*	16.67	13.33	16.67	10.00	6.67	6.67	6.67	16.67	16.67	16.67
*3*	-	-	-	-	-	-	-	-	-	-
*4*	13.33	13.33	33.33	13.33	13.33	16.67	10.00	16.67	33.33	33.33

1. *C. islandica*; 2. *C. islandica*/EAE/AgNPs; 3. *F. esculentum*; 4. *F. esculentum*/EAE/AgNPs. EAE—multienzyme-assisted extract.

## Data Availability

All data generated during this study are included in this article.

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
