# Peer review of "Antimicrobial Activities against Opportunistic Pathogenic Bacteria Using Green Synthesized Silver Nanoparticles in Plant and Lichen Enzyme-Assisted Extracts"

_plants, 2022, doi:10.3390/plants11141833_

Round 1
Reviewer 1 Report
Dear Editor,
once more thank you for the opportunity to review the paper entitled Antimicrobial activities against opportunistic pathogenic bacteria using green synthesized silver nanoparticles in plant and lichen enzyme-assistance extracts. After the significant editing of the previously submitted manuscript the authors have considerably improved the quality of the paper. Therefore, my suggestion is to accept this paper in present form.
Reviewer 2 Report
Thank you very much for all your comments on my suggestions. I am satisfied with these answers. Additionally, the authors made changes to the manuscript regarding my suggestion (Line 21).
Yours faithfully
Reviewer 3 Report
I think the manuscript can be accepted.
This manuscript is a resubmission of an earlier submission. The following is a list of the peer review reports and author responses from that submission.
Round 1
Reviewer 1 Report
The manuscript entitled Antioxidant and antimicrobial activities of novel green synthesis of silver nanoparticles using enzyme-assisted water extraction of plant and lichen is very interesting and well-presented research. The used literature is up to date and correctly chosen. In my opinion this paper should be accepted after minor revision. Some reasons for this statement are given below.
Line 60: please, throughout the text use word lichens for lichens not plants
Line 61: C. islandica should be in italic
Lines 75, 78 and 86: correct the typing mistakes regard mixing numbers and letters
Rename section 2.1. into Lichen and plant material
Line 92: change plants material to lichen material
Line 109: the name of the species should be written with a small letter
Line 162: Change cooled to solid
Line 169: Staphylococcus aureus should be in italic
Figure 1: It would be much easier for the readers if you mark characteristic details on SEM images. Also, change SEM imagines to SEM images.
Figure 3: Explain what indicates the first peak on the EDX spectra because it is not identified on the both specta
Line 258: Change In vitro to in vitro
Table 3: Please, restructure this table, because the units can not be read from it
Table 4: Use bacteria for plural, not bacterias
The conclusion is clear and well-structured, but my advice is to emphasize the novelty and importance of the study in the introduction part.
Author Response
The authors improved the manuscript according to Your and other Reviewers' comments. Authors are grateful for the insights which help authors improve. Below we answered your comments.

Reviewer 2 Report
On request of Plants, I have revised the manuscript titled “Antioxidant and antimicrobial activities of novel green synthesis of silver nanoparticles using enzyme-assisted water extraction of plants and lichen” by Aistė Balčiūnaitienė and co-workers.
Although it is very difficult to understand what the author have done in their study, due to a great confusion and a catastrophic English language, I think that authors have prepared AgNPs by a well-known green method using plants’ extracts. Particularly, they selected the pseudocereal Fagopyrum esculentum (F. esculentum) and the lichen Certaria islandica (C. islandica) as plants and they obtained the extracts by using both a conventional method and a multienzyme-assisted extraction technique, with the scope to compare the AgNPs from both extracts in terms of physicochemical and morphological characteristics, as well as their antibacterial and antioxidant effects.
The argument is not new and much less original. The production of metal nanoparticles (NPs) avoiding the use of dangerous chemical reducing agents has now been extensively studied as well as the practice of obtaining more efficient extracts (in terms of reducing power) through extraction assisted by hydrolytic enzymes. But my biggest concern to this work is the total confusion that characterizes the entire manuscript. English is very bad, very serious grammatical errors, sentences without verbs or subjects, incorrect verbs, incomplete sentences are present in all the work. It is enough just to read the title to understand the bad quality of this manuscript. Ideed, from the title, it seems that the antibacterial and antioxidant activity is of the synthesis and not of the AgNPs. Authors use EXTRACTION in place of EXTRACTS. They erroneously assert that lichens are plants‼
Essential experiments are lacking: the authors do not provide any chromatograms of the extracts they use. The composition of the extracts remains unknown. The FTIR spectra of the extracts are missing and therefore it is not possible to make a comparison with the FTIRs of the nanoparticles obtained using the extracts. The authors go from comparing the properties of the particles obtained with the two types of extracts to comparing instead (at least so it can be deduced) those of the extracts with those of AgNPs (obtained as? It is not always clear). The images of the SEM (Figure 1) are questionable. The caption is not understandable; the authors talk about particles, about what particle? Perhaps the AgNPs obtained using the extracts obtained enzymatically and those obtained with traditional extracts. It is not understandable. Then talking about particles seems absurd to me when looking at the images. TEM images are not compatible with SEM images. It is also absurd to say that the size of the AgNPs is about > 50 nm. So the NPs could be either 50 nm or one meter ???? Where is a Gaussian that shows the particle size distribution? But still, the microbiology data lacks the unit of measurement.
I do not mean that the authors did not work but they worked badly and then wrote worse the results of their already weak work.
The result is that of a manuscript that cannot be published.
Author Response

(The authors gave the same response as above.)

Reviewer 3 Report
In a paper submitted for review entitled "Antioxidant and antimicrobial activities of novel green synthesis of silver nanoparticles using enzyme-assisted water extraction of plants and lichen", the authors based on their own results concluded that tests performed against Gram-positive and Gram-negative bacteria and fungi using disc diffusion or Kirby-Bauer and MIC methods containing different types of extracts showed significant changes. Most of the inhibition zones that were observed confirmed the diffusion of silver nanoparticles from extracts into the medium. The antimicrobial effect of pure extracts has a biocidal effect only in C. islandica. Pure extracts show poor antimicrobial activity. However, AgNPs from C. islandica/EAE exhibited remarkable antibacterial effects that were found to be greater than agNP synthesis from F. esculentum.
The Authors' observations are consistent with previous results of studies by other authors stating that AgNPs have antibacterial properties against: Staphylococcus aureus, Pseudomonas aeruginosa, Xanthomonas axonopodis pv. citri, fungicidal Candida albicans, Candida parapsilosis, as well as antiviral, including for the SARS-CoV-2 virus. Silver nanoparticles also exhibit significant biocompatibility resulting from bidirectional interactions between nanoparticles and host cells or tissues. They also have optical and electronic properties, depending on the size and shape.
The results of the research do not raise my objections and are well documented.
However, I have two doubts and I would like to get information from the Authors:
1. In paragraph 117, the authors state "... under vigorous stirring...". In my opinion, this is an imprecise term. No information about the equipment used and mixing parameters is provided.
2. In paragraphs 139-140, the authors stated that the diameter of the AgNP was measured using the ImageJ-win32 software. The program is not calibrated with TEM. How was the calibration performed when measuring the size of nanoparticles in TEM images?
Yours faithfully,
Author Response

(The authors gave the same response as above.)

Reviewer 4 Report
Review comments on “Antioxidant and antimicrobial activities of novel green synthesis of silver nanoparticles using enzyme-assisted water extraction of plants and lichen”
General comments
This is a research article discussing the application of plant tissue enzyme to synthesis the nanoparticles. The study is well organized; and authors used many methods in materials properties detection. The authors should do a major revision to improve the manuscript quality for our publication. I have several necessary concerns for your study. The detailed comments are listed below,
1. Why Fagopyrum esculentum and lichen Certaria islandica extraction are used for the nanoparticle synthesis.
2. The sem figures showed that the nanoparticles are not in good size, and only part of the particles are in nano size. It is not a good nano concentration for the study.
3. It is not a standard method for nanoparticle research, please confirm your nanoparticles are all in the same size or similar size, how about using zetameter to detect the average size?
4. A experimental blank needed for your comparison, an only Ag NPs and two extraction blanks need for the antibacterial experiment.
5. Figure 3 section, I think the extraction contributes more organic groups rather than the metal elements.
Author Response

(The authors gave the same response as above.)

Round 2
Reviewer 2 Report
Dear Authors,
I have considered your manuscript again. Concerning the author’s explanation provided me in their responses, I note that I have understood what it has been made in this study, the problem is the very confusing way in which the authors have exhibited their work.
Additionally, the confusion is also in their replays, which have been written in a very bad English.
Anyway, the abstract that in my opinion had to be re-written, is practically unchanged.
Virtually none of my requests have been met, the required missing experiments have not been done, the issues raised (see SEM/TEM) remain identical. The only thing that was added following my request is the unit of measurement in microbiology experiments. Unfortunately, by adding this data, the authors have included a big and unacceptable mistake in the work. In fact, the authors gave the mL as the unit of measurement of the MICs. The MIC unit of measurement should be a concentration usually expressed in µg/mL or similar.
The work is not suitable for publication on Plants.
Reviewer 4 Report
I hope the authors do some major revisions, but I only see some words revision. The result needs more data and analysis. Please consider my last comments again. You should do some experiments rather than word revision. It is not good with no reference to support your response letter!